# Neuroprotective Role of Hypothermia in Acute Spinal Cord Injury

**DOI:** 10.3390/biomedicines10010104

**Published:** 2022-01-04

**Authors:** Hasan Al-Nashash, Angelo H. ALL

**Affiliations:** 1Department of Electrical Engineering, College of Engineering, American University of Sharjah, Al Sharjah 46831, United Arab Emirates; hnashash@aus.edu; 2Department of Chemistry, Faculty of Science, Hong Kong Baptist University, Kowloon Tong 999077, Hong Kong

**Keywords:** general hypothermia, local hypothermia, spinal cord injury models, neuroprotection, hypothermia mechanisms of action, hypothermia complications

## Abstract

Even nowadays, the question of whether hypothermia can genuinely be considered therapeutic care for patients with traumatic spinal cord injury (SCI) remains unanswered. Although the mechanisms of hypothermia action are yet to be fully explored, early hypothermia for patients suffering from acute SCI has already been implemented in clinical settings. This article discusses measures for inducing various forms of hypothermia and summarizes several hypotheses describing the likelihood of hypothermia mechanisms of action. We present our objective neuro-electrophysiological results and demonstrate that early hypothermia manifests neuroprotective effects mainly during the first- and second-month post-SCI, depending on the severity of the injury, time of intervening, duration, degree, and modality of inducing hypothermia. Nevertheless, eventually, its beneficial effects gradually but consistently diminish. In addition, we report potential complications and side effects for the administration of general hypothermia with a unique referment to the local hypothermia. We also provide evidence that instead of considering early hypothermia post-SCI a therapeutic approach, it is more a neuroprotective strategy in acute and sub-acute phases of SCI that mostly delay, but not entirely avoid, the natural history of the pathophysiological events. Indeed, the most crucial rationale for inducing early hypothermia is to halt these devastating inflammatory and apoptotic events as early and as much as possible. This, in turn, creates a larger time-window of opportunity for physicians to formulate and administer a well-designed personalized treatment for patients suffering from acute traumatic SCI.

## 1. Introduction

The practical aspects of inducing hypothermia after acute traumatic spinal cord injury (SCI) were adopted many years ago, though its science is not yet fully known. Still, scientists have curious debates whether hypothermia is a myth that gives patients false hope or factual, by which physicians could confidently implement it in their therapeutic approaches. Perhaps hypothermia could become a new standard in treating acute traumatic SCI as one of the potential first treatment steps. To this end, various hypotheses have been investigated to provide definitive answers on the actual short-term and long-term neuroprotective effects of hypothermia [1,2,3,4].

In addition to the severity of the primary injury, a few other critical aspects must be considered thoroughly as well. Indeed, the real benefit of hypothermia is a time-sensitive matter. The overall notion of time-dependency benefits of hypothermia includes (i) patient transportation time (how early procedure can start), (ii) how fast the cooling induction should be, (iii) what the lowest therapeutic temperature should be and for how long, and (iv) how slow the re-warming process (to 37 °C) should be administered [2,5,6,7]. Clinical scientists agree that perhaps hypothermia does not have any significant neuroprotective role in patients with chronic (years) SCI, nor will it be very effective if the hypothermia starts late during the chronic (tertiary) phase of SCI [7,8]. The other critical aspect is how the hypothermia should be administrated, namely: general or local; invasively, minimally invasively, or non-invasively [1,2,5,9]. This is because the hypothesis behind their mechanisms of action and their beneficial effects as well as potential complications are entirely different [10]. Indubitably, the deciding factors for choosing the type of hypothermia induction also depend on other underlying clinical conditions of patients.

Nonetheless, another critical aspect to consider is whether inducing moderate but prolonged (e.g., 31–34 °C for 4–8 h) hypothermia is more effective than a more aggressive reduction in temperature but for a shorter time (e.g., 28–31 °C for 1–3 h). Of course, inducing hypothermia as treatment in SCI patients should be personalized considering the severity of the injury, complications, existent of other pathologies, etc. However, evidence points to the fact that overall mild hypothermia for a short time or severe hypothermia for a longer time would not be preferable. Hence, the aim has often been to induce more moderate but prolonged hypothermia. Moreover, it has also been demonstrated that the early induction of non-invasive moderate hypothermia in patients with traumatic injury (with or without the SCI) will not cause any harm [5,6].

Thus far, nearly all the situations mentioned above have been investigated and been demonstrated that early hypothermia provides the highest neuroprotection post-acute SCI [1,2]. Moreover, it has also been argued that the risk of developing more severe complications is relatively higher in general than the local hypothermia [11,12].

In this paper, we describe alternative forms of hypothermia and discuss various possible hypotheses regarding its mechanisms of action. We also report complications and side effects of general hypothermia and compare it to local hypothermia. In addition, we provide results indicating whether early hypothermia could be considered neuroprotective in acute and sub-acute phases of SCI and for how long. 

## 2. Hypothermia Induction

There are mainly two approaches for inducing hypothermia: general and local, and three modalities: invasively, minimally invasively, and non-invasively. Here, we describe the four major types of hypothermia.

### 2.1. Non-Invasive General Hypothermia

This is a straightforward procedure in which the entire body’s temperature is reduced. Spraying a mixture of an equal volume of 70% alcohol and 10 °C cold water and using a small fan for blowing cool air on the surface of the entire rodent wet body will gradually reduce the temperature [1,5]. There are mainly three steps to induce general hypothermia: safely reducing body temperature by 1 °C every 3–5 min (often from 37 °C to 32 °C), then keeping constant the low temperature for about 120 min (though maintaining it for an even longer time is easily feasible), and finally slow re-warming 1 °C every 5–7 min to 37 °C. A rectal probe measuring core body temperature provides the feedback necessary to manage body temperature reduction by cold mix spray and fan blow. The re-warming, which is a very critical step, is performed by slowly increasing the temperature of the underneath heating-pad. Nevertheless, since the temperature of all organs will be reduced and then re-warmed, some severe complications are expected and, unfortunately, are inevitable. It is also important to mention that, although the core body temperature could be maintained at 32 °C without major variations, the internal organs (spinal cord, brain, etc.) may have deviation up to +/−2 °C.

### 2.2. Invasive General Hypothermia

This method has been used in clinical research settings. The hypothermia is delivered intravascularly via inserting a catheter in the femoral vein. An example of such a catheter is Quattro^®^ Intravascular Heat Exchange Catheter (Custom Luer Model IC4593/8700-0783-01). A critical advantage of this technique is that it can be performed soon after the patient is admitted. The cooling rate is approximately 0.5 °C/h, and the target temperature is often moderate hypothermia at 32–34 °C, which can safely be kept for a long time, even up to 48 h. The re-warming rate to 37 °C is critical and is set very slowly at about 0.1 °C/h [3,13,14]. Although managing potential complications is easier than non-invasive general hypothermia, there will still be inevitable risks and other practical limitations. In addition, the internal organs may have +/−1 °C deviation as well.

### 2.3. Invasive Local Hypothermia

This is possibly the least desired method to induce hypothermia because of the risks and complications of surgical procedures, though it provides a well-controlled temperature reduction restricted to the region of interest [15]. One of the techniques used to induce local hypothermia invasively in rodents is by performing a larger than usual laminectomy at the site of SCI and without opening the dura, continuously delivering drops of cold water to the opening site, and gently aspirating them from the opposite side. By controlling the temperature of the water and the circulation time, reaching, and maintaining the desired temperature is easily feasible for even a long time. However, this is considered not practical in clinical settings because of potentially serious complications such as bleeding, infection, cord compression by the weight of water, etc. This model was primarily used in rodent experiments for the cell, molecular, and histopathological examinations of the spinal cord after hypothermia, and rodents were intended to be euthanized within the 12 h post-SCI.

### 2.4. Minimally Invasive Local Hypothermia

As we described it in detail [9], inducing minimally invasive local hypothermia is easily achievable by using a “heat-exchanger”, which is comprised of a 12 cm length M-shaped 2 mm diameter copper tube and a peristaltic pump [2,5,6,9]. The surgical procedure is minimally invasive and involves a 1 cm transverse incision of the skin at the site of SCI and making a narrow tunnel simply by inserting the index finger under the skin. This model does not require any form of laminectomy. The center of the M-shaped tube is placed over the epicenter of the injury site (just under the skin), and then secured to the skin by small single sutures. Then, the two extremities of the M-shaped tube connected to the peristaltic pump to circulate cold water (17 °C) at the rate of 129 mL/min. By adjusting the water temperature and water circulation speed, any target temperature can easily be reached and maintained for a long time. Remarkably, the system will not make any contact with the spinal cord tissue and could selectively reduce the temperature of only a few underneath spinal cord segments slowly, safely, and non-invasively. Simultaneously, the temperature of the core body, the spinal cord, and paravertebral muscles of the target regions could continuously be monitored via microprobe thermocouples to establish a feedback system for the peristaltic pump function rate and control of the hypothermia temperature precisely. The same heat-exchanger system can also be used to slowly re-warm the spinal cord to 37 °C efficiently. This system allows for the highest precision and complete control of administrating the cooling phase, reaching, and maintaining the target temperature, and the re-warming phase. It carries very minimal risks and is without major complications. It can safely, efficiently, and cost-effectively be implemented in clinical settings as well.

## 3. Early Hypothermia: Potential Mechanisms of Action

Although there are various hypotheses, and each one theoretically provides pieces of evidence for the neuroprotective effect of hypothermia, no consensus regarding hypothermia mechanisms of action has been assented yet. Hypothermia may slow the metabolism, decreasing the need for oxygen and glucose consumption in the tissues. This would reduce the risk of energy failure and in turn, reduces the risk for loss of neural cells. It was also reported that hypothermia would significantly reduce the release of excitotoxic neurotransmitters and the formation of free radicals. Both mechanisms have critical roles in the survival of the neural cells and preservation of the parenchyma. On the other hand, if hypothermia is initiated during the very early acute phase of injury, it may also inhibit pro-inflammatory and apoptotic pathways, consequently decreasing the edema, reducing the inflammation, and lowering blood volume at the injury site. These elements play an essential role in stabilizing the blood–spinal cord barrier and render the desired neuroprotective effects [1,3,4,10,11].

Since electrical conduction (action potential generation) and propagation (sequential depolarization and hyperpolarization) are the main functional entities of the neurons, preventing sustained electrical depolarizations of neuropathways post-injury has also been the focus of many scientists. Hypothermia may prevent sustained depolarizations of under-stress neuropathways that otherwise would become dysfunctional and, in time, could turn non-functional. This mainly happens in spared fibers that survive the immediate injury impact. Spared fibers are anatomically continued axons that pass through the injury site uninterrupted but are not functional. This has been attributed to the post-injury demyelination process. Injury causes a significant reduction in the in situ oligodendrocytes population, disrupting myelin production and thus, halting the normal myelination process. Periodically stimulating these under-stress neuropathways situated within a hostile microenvironment will prevent their dysfunction and ultimately death. Such neural activity has a pivotal role in preserving the neural circuitry network. We have recently demonstrated the effect of various forms of electrical, magnetic, and optogenetic stimulation on neuronal cell survival and remyelination, using in vitro models [16,17,18]. It was also demonstrated that combing electrical stimulation and a long-term physical therapy and rehabilitation regimen would produce a vastly superior outcome in SCI patients [19,20,21,22]. Overall, the protective roles of early hypothermia post-acute SCI could be listed as follows [1,4,10,11]:➢moderating inflammation;➢decreasing the formation of reactive-astrogliosis;➢preserving the blood–spinal cord barrier;➢slowing metabolism;➢decreasing the formation of free radical;➢protecting against acute axonal degeneration;➢inhibiting excitotoxicity;➢preventing sustained electrical depolarization;➢possible neurogenesis.

## 4. General Hypothermia: Potential Complications

Inducing general hypothermia may also cause major and sometimes severe complications if it is not administered and appropriately governed. For instance, hypothermia causes peripheral vasoconstriction. Although one of the desired effects of hypothermia is to slow the perfusion in and around the contused spinal cord parenchyma, this will also affect kidney perfusion and could eventually be turning into renal dysfunction, in which patients may require strict fluid management. Hypothermia may also cause bradycardia and reduced myocardial contractility, adversely affecting the cardiac output and the blood pressure with a potential risk of cardiac fibrillation. Hypothermia may impair leukocyte phagocytic function and influences immuno-suppression as well. In those cases, controlling possible developments of pneumonia and bacterial infections will be the highest priority. It was also reported that hypothermia could trigger mild coagulopathy and platelet dysfunction with serum K^+^, Mg^2+^, and phosphate decreases. Hypothermia could also increase the solubility of the CO_2_ in blood and hence decrease the pCO_2_ and increase the blood pH. Patients during hypothermia may show signs of insulin resistance (hyperglycemia) and exhibit an increased insulin sensitivity (hypoglycemia) during the re-warming period. However, the most critical and life-threatening complication of the general hypothermia would be uncontrollable shivering due to alteration of thermoregulatory defense mechanisms, and if it is not controlled well, it can be fatal [3,4,10,11,12]. Interestingly, local hypothermia is safer and does not cause many of the general hypothermia complications. Moreover, it was also demonstrated that local hypothermia has better outcomes. This was particularly shown in the rodent model of SCI assessed by objective neuro-electrophysiology and motor behavioral examinations (data is presented further) [2,5,6,9].

## 5. Spinal Cord Injury Models

Various reproducible and reliable models of SCI in rodents mimicking the injury in humans with similar anatomopathological characteristics have already been developed. Here, we report four main models used in the basic and translational research.

### 5.1. Contusion

In the contusion model of SCI (e.g., traumas), high energy of impact is delivered to a small area of the spinal cord parenchyma in a very short time, and the impact energy is mostly absorbed by just very few underneath segments. The injury starts from the center Gray matter and appears in the form of the cavities. The size of cavities and the progress of injury to the proximal, distal, ventral, dorsal, and lateral areas are of utmost irregularity and depend on the severity of impact [23,24,25].

### 5.2. Compression

In the compression model of SCI (e.g., tumors), often, there is low pressure for a more extended time directly on the spinal cord tissue. Because of the nature of this injury, smaller nerves are more vulnerable than larger nerves, and the progress of injury is usually much slower than the contusion model [26,27,28].

### 5.3. Transection

In the transection model of SCI (e.g., lacerations), penetrating trauma, similar to a shattered bony structure or sharp object, usually causes fragmentation in axons. This will develop into acute axonal degeneration (AAD) injury, followed by Wallerian degeneration (WD) of the involved surrounding axons. The injury progress is predominantly proximal with limited anatomical extension and severe physiological consequences [29,30,31,32,33].

### 5.4. Focal Demyelination

In the experimental autoimmune encephalomyelitis (EAE) model of SCI (e.g., MS), researchers aim to develop focal demyelination injury along the White matter in the CNS. This injury is well-demarcated (usually less than a cm area) and limited to the target area(s), with little progress over time. In this model, the anatomical structures of axons are essentially preserved, though they are not functional due to the demyelination injury [34,35,36,37,38].

## 6. Neuroelectrophysiology Monitoring

Neuroelectrophysiology is the only “objective” clinical and research tool for the functional assessments of the pathways in the nervous system.

### 6.1. Somatosensory Evoked Potential

Somatosensory Evoked Potential (SSEP) is the electrical activities of the brain used to assess the integrity of ascending sensory neuropathways. It is achieved by electrical stimulation of peripheral nerves (usually Median nerves of the upper limbs and Tibial nerves of the lower limbs) and recording somatosensory signals from the corresponding contralateral cortices. The SSEP measurement is a non-invasive procedure and is performed routinely in clinical settings and in the research models of SCI. It provides objective assessments of the onset, severity, and progress of the injury, as well as the endogenous and therapeutic recovery post-SCI [39,40,41,42,43,44,45,46,47,48,49,50,51]. The SSEP has extensively been used to assess stem cell replacement therapy [52,53,54,55] as well as plasticity and reorganization of neuropathways [31,56] post-SCI as well.

The SSEP has mainly four (millivolt mV) average peaks, and each peak appears at a well-defined (millisecond ms) time. Their most prominent signal components are identifiable as first positive peak P1 (at ~5 ms), first negative peak N1 (at 5–10 ms), second positive peak P2 (at 15–25 ms), and second negative peak N2 (at 25–55 ms) post-stimulation. Any signal before the 5 ms is considered stimulation artifact, and after the 55 ms is part of the EEG signal. For the analysis, the recorded SSEP signal (between 5 and 55 ms) is filtered with a bandpass filter with a bandwidth of 20 Hz to 1K Hz. Ensemble averaging 100 to 700 sweeps is then used to improve the signal-to-noise ratio and extract the noise-free SSEP signals. Peak detection is then applied to locate the N1, and P2 peaks of the averaged SSEPs. The averaged SSEP signals are then normalized relative to the respective or corresponding baseline signal. This is computed by dividing the N1-P2 peak-to-peak amplitude of the SSEP by the N1-P2 peak-to-peak of the corresponding baseline. For SSEP monitoring, often a mixed flow influx of 1.5% isoflurane, 80% oxygen, and room air at a rate of 1.5–2 L/min is used. The rodent’s mouth and nose are placed within an anesthesia mask, and the mask is connected to a C-Pram circuit designed to deliver and evacuate the gas through one tube. The i.p., i.v., or i.m. anesthesia, such as Ketamine cocktails, is not recommended for the SSEP monitoring because they would suppress the brain activities, and, hence, the detection of the SSEP signals would be more difficult.

### 6.2. Motor Evoked Potential

Motor Evoked Potential (MEP) on the other hand, is the electrical signal response used to assess descending motor neuropathways. It is performed by stimulating (electrically, magnetically, optogenetically, etc.) motor cortices and or other higher structures in the nervous system and measuring the evoked activities in the peripheral descending motor pathways and their corresponding muscles. Similarly, the MEP is a non-invasive procedure that can be employed in the clinical settings as well as rodent model of SCI for longitudinal assessments of motor pathways [57,58,59,60]. For the MEP recordings, usually 0.1 ml i.p. injection of Ketamine, Xylazine, and Atropine (7.0-1.0-0.5 cocktail) 15 min prior to monitoring is used and this would be the best choice of anesthesia for MEP monitoring.

## 7. Motor Behavioral Examination

Although various modalities of motor behavior examinations have been developed, the Basso, Beattie, and Bresnahan (BBB) locomotor scale method [61,62] currently is one of the most adopted examinations in evaluating the onset and progress as well as recovery of the SCI in rodents (both in mice and rats). The BBB scoring is rather a subjective evaluation tool and is based on 4 min observation of rat’s locomotion in an open field (90 cm diameter and 30 cm wall plastic pool space) by two examiners standing in front of each other. The BBB assessment consists of three phases of early (0 to 7 score), intermediate (8 to 14 score), and late (15 to 21) recovery scoring, in which 0 means no joint or limb movements and 21 means no locomotion deficit [1,23,29,30,34,63].

It is noteworthy to briefly mention here that although defining the role of hypothermia [64] in protecting the CNS either post-trauma or during surgical procedures (such as oblique corpectomy tumor resection) [65] is crucial, other approaches, such as omega-3 fatty acids [66] treatment post-SCI should also be investigated.

## 8. Materials and Methods

In our studies, we used the NYU-Impactor device, which is a well-established SCI model, to induce moderate contusive (12.5 mm) SCI at T8 in adult (250 g) male and female rats. Rats were under general gas anesthesia (1.5–2% isoflurane, 80% oxygen, and room air delivered by the rodent vaporizer and ventilator) during all the surgical procedures and hypothermia induction phases.

The treatment of rats was strictly according to the guidelines set by our university Institutional Animal Care and Use Committee (IACUC), Neuroscience Research, NIH Guidelines for the Care and Use of Laboratory Animals, and as per our extensive publications.

Here, we report the results for (i) non-invasive 2 h general hypothermia by spraying a mixture of alcohol/cold water and use of a small fun with feedback from core body temperate measured by a rectal probe and (ii) 5 h and (iii) 8 h minimally invasive local hypothermia using M-shaped heath-exchanger copper tube placed under the skin above the epicenter of SCI. The rats in the control group had identical anesthesia drug, anesthesia time, and surgical procedures, but they underwent only laminectomy (without any injury) at T8 and had no body temperature manipulation. Rats’ longitudinal neuroelectrophysiology assessments were completed under general anesthesia, and the SSEP signals were analyzed offline. Motor behavioral (BBB scoring) examinations were conducted in awake and mobile rats. At the end of experiments, after inducing general anesthesia deeply, rats were euthanized according to the NIH Guidelines. 

## 9. Results

### 9.1. Inducing Hypothermia is not Harmful

Our results demonstrated that inducing local hypothermia, even aggressively for a relatively long time, neither cause any adverse effect on the spinal cord or surrounding parenchyma, nor on other parts of the nervous system. Healthy adult male and female rats (*n* = 15) were subjected to 30 °C ± 0.5 °C local hypothermia at T7–T9 for 5 h and 8 h, while their core body temperature was maintained at 37 °C ± 0.5 °C. Figure 1a,b show the amplitude of SSEPs from control, 5 h, and 8 h local hypothermia recorded when stimulating left and right hindlimbs, respectively. Figure 1c shows the average BBB scores of the same cohort of rats. Four weeks of longitudinal neuro-electrophysiology SSEP monitoring, and motor behavior BBB examinations showed absolutely no functional deterioration or locomotion deficit.

### 9.2. The Temperature Profile of Local and General Hypothermia

We monitored the profile of temperature changes at the following three sites: spinal cord injury epicenter, cortex (considering the role of tissue continuity and blood circulation on the CNS), and the core body of adult rats (*n* = 15) that underwent moderate (12.5 mm NYU-Impactor) contusive SCI. Rats were subjected to either local 30 °C ± 0.5 °C hypothermia for 5 and 8 h or general 32 °C ± 0.5 °C hypothermia for 2 h and compared to the control (laminectomy without SCI) normothermia 37 °C ± 0.5 °C group. Practically, during the general hypothermia, the core temperature of rats was successfully reduced to and maintained at 32.1 °C ± 0.4 °C, and during the local hypothermia, the core temperature was successfully maintained at 37.3 °C ± 0.7 °C. As expected, general hypothermia causes a significant reduction in the cortical temperature compared to the effect of hypothermia induction in the spinal cord locally (Figure 2).

### 9.3. Hypothermia is Neuroprotective, Though Temporarily

We investigated the neuroprotective effect of 2 h general hypothermia at 32 °C ± 0.5 °C as well as 5 h and 8 h prolonged local T7–T9 hypothermia at 30 °C ± 0.5 °C using the contusive moderate (12.5 mm) T8 thoracic SCI. To reflect the real-life situations, inducing hypothermia was delayed for 2 h post-SCI induction.

The SSEP results show that the neuroelectrophysiology (SSEPs) improvements of 2 h general, as well as 5 h and 8 h local hypothermia are statistically significant. This, in turn, translates into the neuroprotective beneficial effects of hypothermia. Yet, in all three instances, the improvements were temporary, as depicted in Figure 3a. Similarly to the 2 h general hypothermia, the 5 h local hypothermia exhibited the highest neuroprotective effect early, with longer and more prominent efficacy through the 8 weeks of observation. On the other hand, although the 8 h local hypothermia neuroprotective efficacy remained substantial, it still reduced significantly over the observational time. A similar trend was also noted for the motor behavior examinations and the BBB scores as shown in Figure 3b.

It is important to mention that although the relative amplitude variations in SSEPs between the different hypothermia procedures can reach between 20% to 40%, these changes do not reflect the same level of improved phenotype outcomes. This is observed in the corresponding BBB score values as well. For example, on Day 14, the difference between SSEPs in normothermia and the 2 h general hypothermia is almost 20%. However, the BBB score shows a difference of no more than 7%.

## 10. Discussion

Hypothermia treatment as a therapeutic strategy for acute traumatic SCI still brings passionate debates. Scientists continuously evaluate the potential benefits of hypothermia and the characteristics of its various forms of administration. Moreover, the lack of convincing evidence about the probable mechanisms of its action should also be acknowledged. We presented a review of our results to report that early hypothermia is neuroprotective, and its beneficial effects are prominent during the acute and sub-acute primary phase of contusive SCI. Depending on the severity of the injury, such as in the case of moderate injury, its neuroprotection effect could also extend to the secondary phase, but eventually, it fades. Thus, the outcomes of hypothermia induction post-acute moderate and severe SCI (but not necessarily mild and very severe) are very much promising when early results are being considered and analyzed. Yet, analyzing of few weeks later results may not reveal many of the statistically significant differences. Hence, hypothermia should be considered a palliative treatment rather than an actual therapeutic strategy in patients suffering from SCI. In addition, inducing hypothermia may not show neuroprotective effects for mild SCI due to the excellent endogenous recovery or for very severe SCI because of massive destruction of the anatomical structures of the spinal cord. Indeed, hypothermia does not treat the underlying pathological events but would expand and create a larger time-window opportunity for physicians to execute their treatment strategies and rehabilitation therapists to manage their personalized physical therapies for such a staggering time-sensitive disease. In fact, if it is adequately administrated, moderate and local hypothermia has no risk or complications, nor it is harmful. An example is in the case of SCI patients who would be considered candidates for the stem cell replacement trials. Due to the very hostile microenvironment of the spinal cord parenchyma during the primary and secondary phases of SCI, the cells will not likely survive beyond a few days post-transplantation. Early hypothermia not only could prevent some of the destructive inflammatory processes and spare more neuropathways and parenchyma, but also it could improve the microenvironment where the stem cells are intended to be transplanted. Either way, the improvements in functional outcomes during the primary and secondary phases of SCI will be enhanced. Furthermore, considering a different therapeutic strategy, combining early hypothermia with the long-standing rehabilitation regime will undoubtedly improve the quality of life in patients with SCI.

We conclude that hypothermia is neuroprotective and, if applied early, could limit the secondary phase of traumatic spinal cord injury. However, alone, it cannot be considered a solo treatment for SCI per se. We presume the disagreement about the neuroprotective effect of hypothermia among scientists is rooted in whether their conclusion is drawn from early or late results. The early results show that hypothermia is neuroprotective, while for the late findings, the evidence for neuroprotective beneficial effects is not as strong.

## Figures and Tables

**Figure 1 biomedicines-10-00104-f001:**
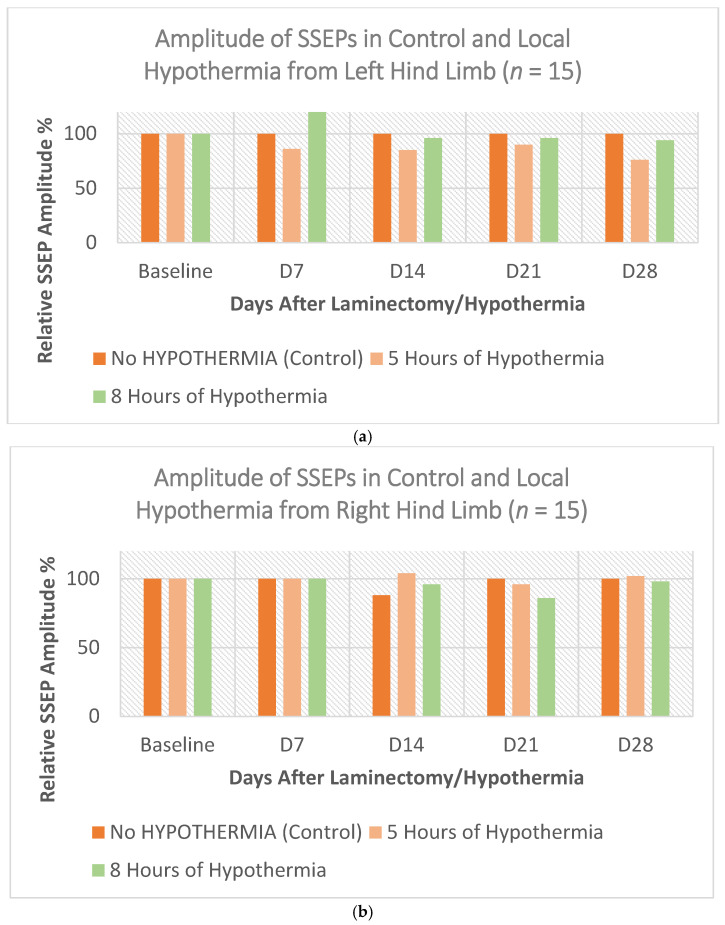
Shows that prolonged local hypothermia is not harmful (*n* = 15). The three plots show SSEP signals of the left hindlimbs (**a**) and the right hindlimbs (**b**) as well as the average BBB scores (**c**) of rats in normothermia, 5 h, and 8 h local hypothermia groups.

**Figure 2 biomedicines-10-00104-f002:**
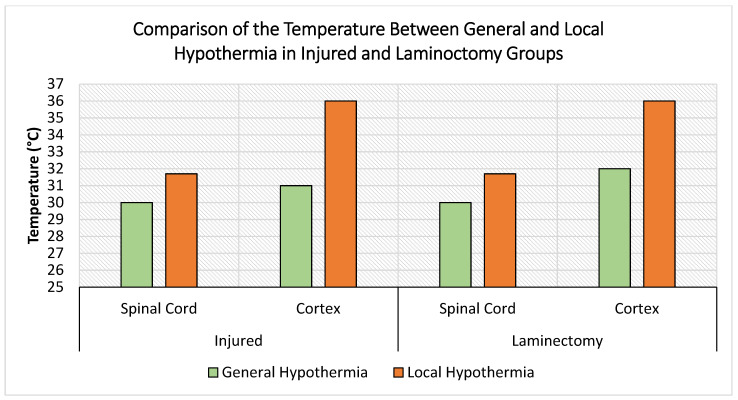
Shows the temperature profiles of rats’ cortex and spinal cord parenchyma after induction of general (32 ± 0.5 °C) and local (30 ± 0.5 °C) hypothermia in laminectomy (control without SCI) on the right and the injury (moderate contusive SCI) groups on the left. During the local hypothermia, the core temperature was successfully maintained at 37.3 °C ± 0.7 °C.

**Figure 3 biomedicines-10-00104-f003:**
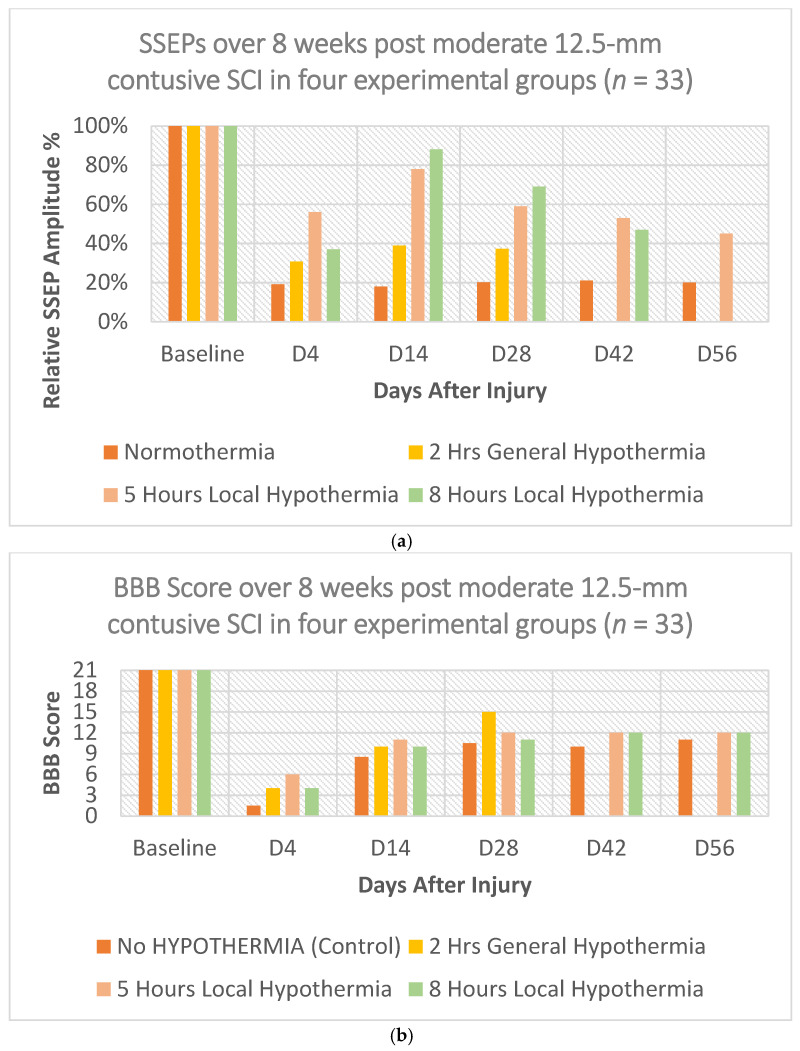
Shows the (**a**) SSEP results and (**b**) BBB scores of normothermia group and the 2 h general (32 °C ± 0.5 °C) as well as 5 h and 8 h local hypothermia (30 °C ± 0.5 °C) in rats with either laminectomy (no injury) or moderate T8 contusive SCI.

## Data Availability

The data of this study are available from the corresponding authors, Al-Nashash and ALL, upon request.

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
