# Peer review of "Neuroprotective Role of Hypothermia in Acute Spinal Cord Injury"

_biomedicines, 2022, doi:10.3390/biomedicines10010104_

Round 1

Reviewer 1 Report

The original article by Al-Nashash et al. "Neuroprotective Role of Hypothermia in Acute Spinal Cord Injury" covers a potentially interesting and emerging topic related to the SCI therapy. In this sense, this remains to be potentially interesting for the Biomedicine readers. I regard the main point of this paper as highly attractive as well as the results are clearly presented. The text does not contain any major errors, therefore I have some minor comments and recommendations:

1.The figure summarizing and clarifying the role of hypothermia in SCI should be added.

2. Table summarizing hypothermia rodent models should be added

3.  Following references should be added and properly cited within the main text:

  • Tykocki T, Poniatowski ŁA, Czyz M, Wynne-Jones G. Oblique corpectomy in the cervical spine. Spinal Cord. 2018 May;56(5):426-435. doi: 10.1038/s41393-017-0008-4.
  • Ahmad FU, Wang MY, Levi AD. Hypothermia for acute spinal cord injury--a review. World Neurosurg. 2014 Jul-Aug;82(1-2):207-14. doi: 10.1016/j.wneu.2013.01.008.
  • Wojdasiewicz P, Poniatowski ŁA, Turczyn P, Frasuńska J, Paradowska-Gorycka A, Tarnacka B. Significance of Omega-3 Fatty Acids in the Prophylaxis and Treatment after Spinal Cord Injury in Rodent Models. Mediators Inflamm. 2020 Jul 29;2020:3164260. doi: 10.1155/2020/3164260.

4.In some places the use of English could be improved on.

Completing this gaps will have an impact on the understanding the aim of the study and, from my point of view, is absolutely necessary.

Author Response

Reviewer #1:

The original article by Al-Nashash et al. "Neuroprotective Role of Hypothermia in Acute Spinal Cord Injury" covers a potentially interesting and emerging topic related to the SCI therapy. In this sense, this remains to be potentially interesting for the Biomedicine readers. I regard the main point of this paper as highly attractive as well as the results are clearly presented. The text does not contain any major errors.

à The authors would like to thank this reviewer for reviewing our manuscript and providing us with excellent comments and suggestions. We have carefully addressed all the comments, and suggestions raised by this reviewer. We have made changes to the manuscript and highlighted in yellow. We do appreciate your effort and think that these comments did improve the overall quality of our work and its presentation.

Therefore I have some minor comments and recommendations:

  1. The figure summarizing and clarifying the role of hypothermia in SCI should be added.

à Please see page 6 highlighted yellow section.

  1. Table summarizing hypothermia rodent models should be added

à Please see page 10 highlighted yellow section.

  1. Following references should be added and properly cited within the main text:

- Tykocki T, Poniatowski ŁA, Czyz M, Wynne-Jones G. Oblique corpectomy in the cervical spine. Spinal Cord. 2018 May;56(5):426-435. doi: 10.1038/s41393-017-0008-4.

- Ahmad FU, Wang MY, Levi AD. Hypothermia for acute spinal cord injury--a review. World Neurosurg. 2014 Jul-Aug;82(1-2):207-14. doi: 10.1016/j.wneu.2013.01.008.

- Wojdasiewicz P, Poniatowski ŁA, Turczyn P, Frasuńska J, Paradowska-Gorycka A, Tarnacka B. Significance of Omega-3 Fatty Acids in the Prophylaxis and Treatment after Spinal Cord Injury in Rodent Models. Mediators Inflamm. 2020 Jul 29;2020:3164260. doi: 10.1155/2020/3164260.

à please see page 9 and page 21 highlighted yellow sections.

  1. In some places the use of English could be improved on.

à Please note that we have reviewed carefully again and revised the manuscript with some minor corrections.

Completing these gaps will have an impact on the understanding the aim of the study and, from my point of view, is absolutely necessary.

à Thank you.

Reviewer 2 Report

The authors provide an overview on the role of hypotermia in the therapuetic managment of patients with traumatic spinal cord injury, focusing on various form to induce hypotermia and hypotesizing different mechanisms of action. Some experimental data are also presented in the form of neuro-electrophysiological results highlighting neuroprotective effects within the first 2 months from injury. The manuscript is timely, the topic is of interest and the study is scientifically sound. I would be in favour of publication.

Author Response

Reviewer #2:

The authors provide an overview on the role of hypotermia in the therapuetic managment of patients with traumatic spinal cord injury, focusing on various form to induce hypotermia and hypotesizing different mechanisms of action. Some experimental data are also presented in the form of neuro-electrophysiological results highlighting neuroprotective effects within the first 2 months from injury. The manuscript is timely, the topic is of interest and the study is scientifically sound. I would be in favour of publication.

à The authors would like to thank this reviewer for reviewing our manuscript and for the positive remarks and comments.